# Drug targeting Nsp1-ribosomal complex shows antiviral activity against SARS-CoV-2

**Mohammad Afsar[1], Rohan Narayan[2], Md Noor Akhtar[3], Deepakash Das[1], Huma Rahil[1], Santhosh Kambaiah Nagaraj[2], Sandeep M Eswarappa[3], Shashank Tripathi[2], Tanweer Hussain[1]***

[1]Department of Molecular Reproduction, Development and Genetics, Indian Institute of Science, Bangalore, India; [2]Microbiology & Cell Biology Department, Centre for Infectious Disease Research, Indian Institute of Science, Bengaluru, India; [3]Department of Biochemistry, Indian Institute of Science, Bangalore, India

**Abstract** The SARS-CoV-2 non-structural protein 1 (Nsp1) contains an N-terminal domain and C-terminal helices connected by a short linker region. The C-terminal helices of Nsp1 (Nsp1-C-ter) from SARS-CoV-2 bind in the mRNA entry channel of the 40S ribosomal subunit and blocks mRNA entry, thereby shutting down host protein synthesis. Nsp1 suppresses host immune function and is vital for viral replication. Hence, Nsp1 appears to be an attractive target for therapeutics. In this study, we have in silico screened Food and Drug Administration (FDA)-approved drugs against Nsp1-C-ter. Among the top hits obtained, montelukast sodium hydrate binds to Nsp1 with a binding affinity ($K_D$) of 10.8 ± 0.2 μM in vitro. *It forms a stable complex with* Nsp1-C-ter in simulation runs *with –95.8 ± 13.3 kJ/mol binding energy*. Montelukast sodium hydrate also rescues the inhibitory effect of Nsp1 in host protein synthesis, as demonstrated by the expression of firefly luciferase reporter gene in cells. Importantly, it shows antiviral activity against SARS-CoV-2 with reduced viral replication in HEK cells expressing ACE2 and Vero-E6 cells. We, therefore, propose montelukast sodium hydrate can be used as a lead molecule to design potent inhibitors to help combat SARS-CoV-2 infection.

**\*For correspondence:**
hussain@iisc.ac.in

**Competing interest:** The authors declare that no competing interests exist.

## Editor's evaluation

This study reports on the repurposing of Montelukast, an FDA-approved drug, with Nsp-1. The Non-structural protein (Nsp)-1 from SARS-CoV2 mimics the binding mode of eukaryotic initiation factor 3 (eIF3j) to the mRNA entry tunnel of the 40S ribosomal subunits and blocks the entry of mRNA, which shuts down host protein synthesis. As a result, the host immune function is suppressed. This makes Nsp-1 an attractive target for therapeutic intervention. In the revised manuscript the authors' claims are supported using Biophysical and cellular assays. They also rationalize their findings using molecular dynamics simulations.

## Introduction

SARS-CoV-2, the causative agent of severe coronavirus disease-19 (COVID-19) pandemic, is an enveloped positive-strand RNA-containing virus and belongs to beta coronavirus family (*V'kovski et al., 2021*). The virus contains nearly 30 kb RNA genome with 5'-cap and 3' poly-A tail (*Finkel et al., 2021*; *V'kovski et al., 2021*). The SARS-CoV-2 genome encodes for 14 open reading frames (ORFs). Upon entry into host cells, ORF1a and ORF1b encode for two polyproteins, which are later

auto-proteolytically cleaved into 16 proteins, namely Nsp1–Nsp16. Among these proteins, Nsp1 binds in the mRNA entry channel of the 40S ribosomal subunit and blocks the entry of mRNAs, thereby shutting down host protein synthesis. Nsp1 also induces endonucleolytic cleavage of host RNAs.

The cryo-electron microscopy (cryo-EM) structures of ribosomes from Nsp1-transfected human HEK293T cells indicate the binding of Nsp1 with 40S and 80S ribosomal subunits (*Schubert et al., 2020*; *Thoms et al., 2020*; *Tidu et al., 2020*; *Vankadari et al., 2020*; *Figure 1—figure supplement 1A*). Nsp1 contains 180 amino acids with N-terminal (1–127 amino acids) and C-terminal (148–180 amino acids) structured regions connected by a loop region of about 20 amino acids (*Schubert et al., 2020*; *Thoms et al., 2020*; *Figure 1—figure supplement 1B*). This C-terminal region of Nsp1 (Nsp1-C-ter) contains two helices that harbors a conserved positively charged motif (KH-$X_5$-R/Y/Q-$X_4$-R). The deposition of positive charge toward one edge of these helices enhances their ability to bind helix h18 of 18S rRNA. The other side of C-terminal helices interacts with ribosomal proteins uS3 and uS5 in mRNA entry tunnel of the 40S (*Schubert et al., 2020*; *Thoms et al., 2020*; *Figure 1—figure supplement 1A*, zoomed view). These interactions enable Nsp1-C-ter to bind deep into the mRNA entry tunnel and prevent the binding of mRNAs, thereby inhibiting host protein synthesis (*Schubert et al., 2020*; *Thoms et al., 2020*; *Tidu et al., 2020*). Thus, Nsp1 helps in hijacking the host translational machinery (*Yuan et al., 2020*) and renders the cells incapable of mounting an innate immune response to counter the viral infection (*Narayanan et al., 2008*). Mutating the positively charged residues K164 and H165 in Nsp1-C-ter to alanines leads to a decrease in binding affinity of Nsp1 with ribosome and fails to inhibit host protein synthesis (*Schubert et al., 2020*; *Thoms et al., 2020*; *Tidu et al., 2020*).

Nsp1 is a highly conserved protein and less than 3% of SARS-CoV-2 genomic sequences analyzed showed mutation in Nsp1 (*Min et al., 2020*). Further, Nsp1-C-ter showed a much reduced frequency of mutations (*Min et al., 2020*). The crucial role of Nsp1 in inhibiting host gene expression, suppression of host immune response (*Thoms et al., 2020*) and, notably, the reduced mutation frequency in Nsp1-C-ter across global SARS-CoV-2 genomes (*Min et al., 2020*) advocate targeting Nsp1 for therapeutics. In this study, we have employed computational, biophysical, in vitro, and mammalian cell line based studies to identify FDA-approved drugs targeting Nsp1-C-ter and check for its antiviral activity.

## Results

Since repurposing a drug is a quicker way to identify an effective treatment, we screened FDA-approved drugs against Nsp1-C-ter (148–180 amino acids) which binds in the mRNA channel (*Figure 1—figure supplement 1C*). The drugs docked to a small region of Nsp1-C-ter consisting of residues (P153, F157, N160, K164, H165, and R171) which coincides with its ribosome-binding interface (*Figure 1—figure supplement 1C*). The residues in Nsp1-C-ter involved in binding drugs show minimal mutations in worldwide deposited 4,440,705 sequences of SARS-CoV-2 genome in GISAID database (*Figure 1—figure supplement 1D*). We identified top hits with at least three hydrogen bonds;(H-bonds) near the ribosome binding site of Nsp1-C-ter (*Supplementary file 1*). Further, the clash that the drugs may have against ribosome in its bound form with Nsp1-C-ter was also analyzed. Montelukast sodium hydrate (hereafter referred to as montelukast) and saquinavir mesylate (hereafter referred to as saquinavir) showed high clash scores (*Supplementary file 1*). Montelukast is regularly used to make breathing easier in asthma (*Paggiaro and Bacci, 2011*), while saquinavir is an anti-retroviral drug used in the treatment of human immunodeficiency virus (HIV) (*Khan et al., 2021*).

Next, all 12 drugs were tested in vitro for their ability to bind to Nsp1. *The purified proteins*, i.e., full-length Nsp1 and C-terminal helices truncated Nsp1 (Nsp1ΔC) proteins, were loaded on the Ni-NTA sensors in bio-layer interferometry (BLI), and the compounds were screened to determine its binding to these proteins. We found that montelukast and saquinavir show binding to Nsp1 (*Figure 1A*) but not with Nsp1ΔC (*Figure 1B*). This indicates that montelukast and saquinavir bind to Nsp1-C-ter. The rest of the compounds does not show binding with Nsp1 or with Nsp1ΔC (*Figure 1A, B*). We next determined binding affinities of montelukast and saquinavir against Nsp1. Montelukast shows a binding affinity ($K_d$) of 10.8 ± 0.2 μM (*Figure 1C*) while saquinavir shows a binding affinity of 7.5 ± 0.5 μM toward Nsp1-C-ter (*Figure 1D*).

To further validate the binding of ligands with Nsp1-C-ter, we performed NanoDSF experiments where we observed the change in the melting temperature of Nsp1 in the presence of drugs. We observed that only montelukast and saquinavir induce a change in the melting temperature of Nsp1 (*Figure 1—figure supplement 1E*). None of the ligands were able to change the melting temperature

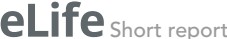

**Figure 1.** Screening and binding kinetics and molecular simulation dynamics runs of drugs against C-terminal helices of Nsp1 (Nsp1-C-ter). (**A and B**) Bio-layer interferometry (BLI) analysis for the initial screening of binding of the drugs with the (**A**) non-structural protein 1 (Nsp1) and (**B**) C-terminal helices truncated Nsp1 (Nsp1ΔC) proteins. (**C and D**) The kinetic behaviors of (**C**) montelukast and (**D**) saquinavir monitored using BLI by incubating increasing concentration of the drug molecule (0–25 μM) on the protein-bound sensors. Montelukast shows a binding constant ($K_D$) of 10.8 ± 0.8 μM, while saquinavir binds with Nsp1-C-ter with a $K_D$ value of 7.5 ± 0.5 μM. (Error bars represent standard deviation of three replicates in (**C**) and (**D**). (**E and F**) Nanoscale differential scanning fluorometry (NanoDSF) experiments to evaluate the change in the melting temperature of the Nsp1 by incubating increasing concentration of (**E**) montelukast and (**F**) saquinavir. (The experiments were performed in three replicates.)) (**G**) Simulation runs with montelukast show stable root mean square deviation (RMSD) values for all replica throughout all molecular dynamic simulation trajectories for 500 ns. (**H**) The analysis of binding mode of montelukast at the end of 500 ns shows stable binding with C-terminal helices. The residues E148 and L149 form H-bonds with montelukast, while F157 and L173 forms base stacking interactions. (**I**) Simulation runs with saquinavir show stable pattern in RMSD values throughout in all molecular dynamic simulation trajectories for 500ns. (**J**) The analysis of binding mode of saquinavir at the end of 500 ns shows stable binding with the C-terminal helices. The residues T151, M174, and R175 form H-bonds with saquinavir, while R171 forms base stacking interactions.

The online version of this article includes the following figure supplement(s) for figure 1:

**Figure supplement 1.** Screening of Food and Drug Administration (FDA)-approved drugs against Nsp1 from SARS-CoV-2 and nanoscale differential scanning fluorometry (NanoDSF) experiments to evaluate the binding of top hits with the non-structural protein 1 (Nsp1) and C-terminal helices truncated Nsp1 (Nsp1ΔC).

**Figure supplement 2.** Structural dynamics of drug-bound complexes of C-terminal helices of Nsp1 (Nsp1-C-ter).

of the Nsp1ΔC protein (*Figure 1—figure supplement 1F*). Next, we performed NanoDSF experiments with different concentrations of montelukast and saquinavir to determine the change in melting temperature of Nsp1. We observed that montelukast shifts the $\Delta T_m$ by 4.3°C while the saquinavir causes a $\Delta T_m$ shift by 6.5°C (*Figure 1E, F*). Overall, montelukast and saquinavir showed binding to Nsp1-C-ter in vitro.

To gain insights into the binding mode of montelukast and saquinavir with Nsp1-C-ter, we analyzed the docked drugs and performed molecular dynamic simulation runs. The molecular screening experiment shows the binding of montelukast with Nsp1-C-ter with a 5.61 docking score (*Supplementary*

*file 1* and *Figure 1—figure supplement 2A*). In the simulation runs the root mean square deviation (RMSD) of C-terminal helices bound with montelukast shows less deviation from the mean atomic positions (*Figure 1G*). The analysis of H-bonds and hydrophobic interactions indicate strong binding of montelukast during the simulation run. At the end of the simulation run, montelukast shows a stable complex by forming H-bonds with E148 and L149, while F157 and L173 form base stacking interactions (*Figure 1H*). The root mean square fluctuation (RMSF) plot shows the thermal stability of individual residues throughout the molecular dynamics run of the molecule, and it appears to be stable (*Figure 1—figure supplement 2B*). Saquinavir shows binding with Nsp1 with a docking score of 5.6 (*Supplementary file 1* and *Figure 1—figure supplement 2C*). The RMSD plot of saquinavir bound C-terminal helices shows reduced deviation of the protein atoms during the simulation runs from the mean atomic position (*Figure 1I*). The residues T151, M174, and R175 form H-bonds with saquinavir while R171 forms base stacking interaction at the end of the run (*Figure 1J*). The RMSF plot show that the participating residues is also stabilized upon the binding of saquinavir (*Figure 1—figure supplement 2D*). Overall, the residues involved in binding montelukast and saquinavir show extremely low mutational frequency.

Furthermore, these drug-Nsp1 complexes were subjected to free binding energy calculations using end state free binding energy for 500 ns in two replicas for each complex. Montelukast and saquinavir bind with Nsp1 with binding energies of –95.8 ± 13.3 kJ/mol and –42.7 ± 5.2 kJ/mol, respectively. The average H-bonds were analyzed for the C-terminal region of Nsp1 alone and drug-bound complexes. We observed that these drugs-bound complexes show higher average H-bonds throughout different replica simulations (*Figure 1—figure supplement 2E*).

Since *Nsp1* is known to inhibit host protein synthesis by blocking the mRNA entry tunnel on the ribosome and co-transfection of Nsp1 with capped luciferase reporter mRNA causes reduction of luciferase expression (*Thoms et al., 2020*). We hypothesized that binding of montelukast or saquinavir to Nsp1-C-ter may prevent inhibition of host protein synthesis. To test this hypothesis, we carried out the cell-based translational rescue of luciferase activity in the presence of montelukast and saquinavir in HEK293 cells when co-transfected with Nsp1. Co-transfection of Nsp1 decreased the luciferase activity by almost half, which is restored by the increasing amount of montelukast (*Figure 2A*). However, we do not observe a similar rescue of luciferase activity in the presence of saquinavir (*Figure 2B*). Further experiments are needed to figure out why saquinavir is unable to rescue the Nsp1-mediated translation inhibition. There was no significant change in gene expression of the firefly luciferase *FLuc* gene (*Figure 2C, D*).

To test antiviral effects of montelukast and saquinavir against SARS-CoV-2, we first tested the cytotoxicity of these drugs in HEK293T-ACE2 and Vero-E6 cells. Results showed minimal toxicity up to 10 μM montelukast and saquinavir in both cell lines. However, in Vero-E6 cells, the highest concentration (20 μM) of both drugs showed an almost 80% decrease in cell viability, compared to untreated cell control (*Figure 3—figure supplement 1*). Based on this, a working concentration of 10 μM or lower was used for both drugs. At a concentration of 10 μM, montelukast showed significant antiviral activity, as indicated by reduced expression of viral spike protein in HEK293T-ACE2 and Vero-E6 cells (*Figure 3A, D*). The corresponding qRT-PCR data demonstrated up to 1-log reduction in viral copy number in both HEK293T-ACE2 and Vero-E6 cells at this concentration (*Figure 3B, E*), supported by a decrease in infectious virus titer measured by plaque assay (*Figure 3C, F*). No significant antiviral effects were observed in the presence of 10 μM saquinavir (*Figure 3—figure supplement 2*).

## Discussion

Nsp1 is a major virulence factor in SARS-CoV2 which effectively blocks the synthesis of major immune effectors (IFN-β, IFN-$\lambda$1, and interleukin-8, retinoic acid–inducible gene I), thereby aiding in establishment of the viral infection (*Thoms et al., 2020*). It serves as a blockage to host mRNA entry by interacting with rRNA helix 18 and ribosomal proteins-uS5 and uS3 near the mRNA entry channel of the 40S ribosomal subunit via its C-terminal helices (*Thoms et al., 2020*). Structural studies on 48S-like preinitiation complex on Cricket paralysis viral internal ribosomal entry site in presence of Nsp1 revealed its ability to lock the head domain of 40S ribosome in a closed conformation. In addition, it competes with eIF3j for uS3 and weakens the binding of the eIF3 to the 40S subunit (*Yuan et al., 2020*). While the host translation is inhibited by the C-terminal helices of Nsp1, its N-terminal domain enhances translation of viral mRNAs by binding to the 5' UTR (*Gordon et al., 2020*). Moreover, Nsp1

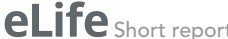

**Figure 2.** Translational rescue experiments in the presence of montelukast and saquinavir. (**A**) Luciferase-based reporter assay shows translational rescue of luciferase in the presence of montelukast. (**B**) Luciferase-based reporter assay shows that saquinavir could not rescue the luciferase expression. Error bars represent standard deviation of three replicates in (**A**) and (**B**). (**C and D**) The real-time PCR to quantitate the fold change of *FLuc* gene in comparison to *GAPDH* in the presence of different concentration of the drug molecules. (**A**) montelukast (**B**) saquinavir. The panel below provides the details of experimental conditions. Error bars represent standard deviation of three replicates in (**A**) and (**B**). The significance of the data was monitored by applying the unpaired t-test through assuming Gaussian distribution parametric test by defining the statistical significance. **p < 0.01; ***p < 0.001; ****p < 0.0001. The error bars represent the standard deviation.

interacts with host mRNA export receptor NXF1-NXT1 heterodimer and aids in retention of cellular mRNAs in the nucleus (*Zhang et al., 2021*). Further, *Mou et al., 2021* deciphered the frequency of mutation accumulation in the N-terminal domain was higher than that of the C-terminal domain (*Mou et al., 2021*). Therefore, we targeted the C-terminal helices of Nsp for this study.

Since repurposing a drug is a quicker way to identify an effective treatment, we screened FDA-approved drugs against Nsp1-C-ter and found montelukast as potential lead molecule against it. Montelukast is a leukotriene receptor antagonist and repurposing montelukast for tackling cytokine storms in COVID-19 patients has been suggested (*Sanghai and Tranmer, 2020*) and hospitalized COVID-19 patients that were given montelukast had significantly fewer events of clinical deterioration (*Khan et al., 2021*). Montelukast also appears as a hit against the SARS-CoV-2 main protease, (M<sup>pro</sup>)

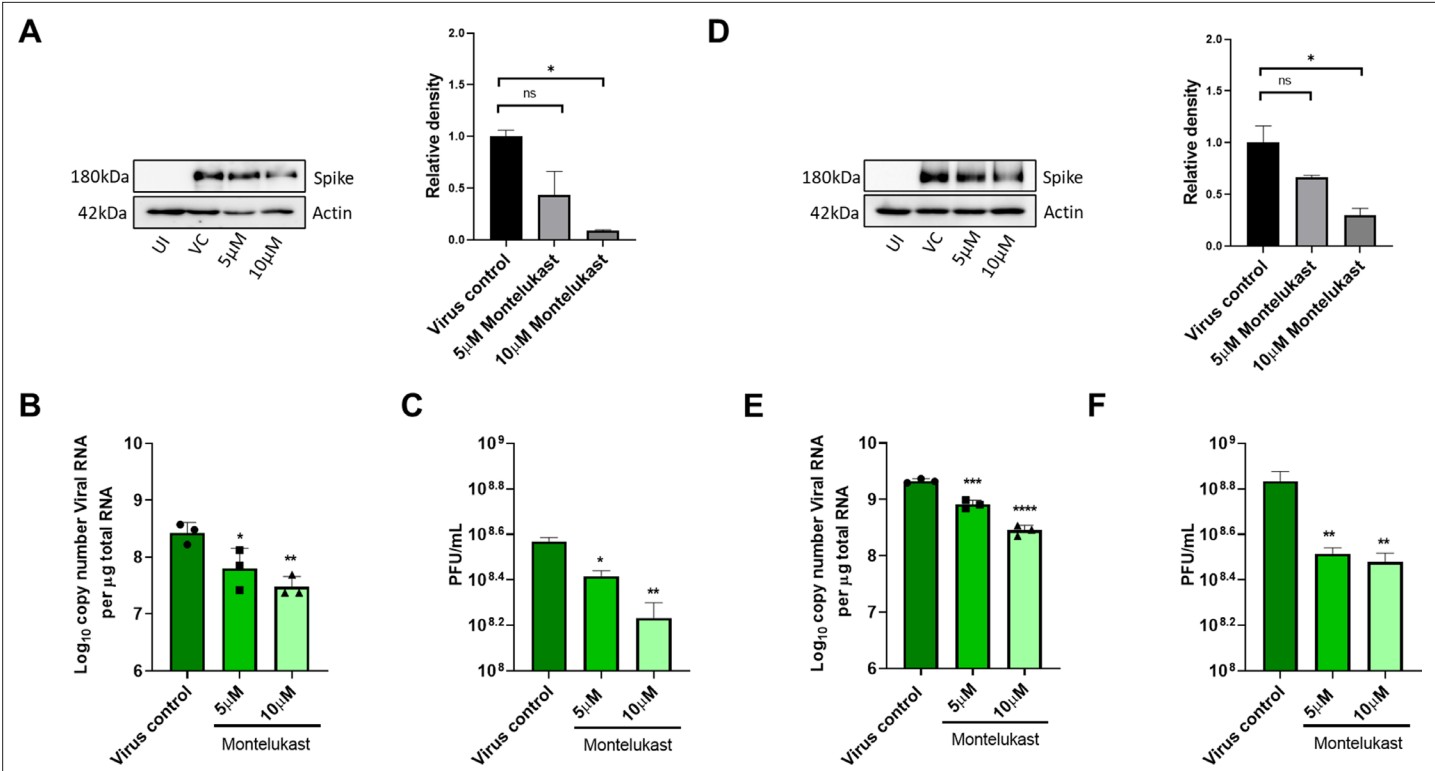

**Figure 3.** Montelukast shows antiviral activity against SARS-CoV-2. (**A**) HEK ACE2 cells were pre-treated with 5 or 10 µM montelukast and infected with 0.1 MOI SARS-CoV-2 for 48 hr. Virus spike protein expression by western blot analysis, with corresponding relative density of bands are shown. (**B**) Viral RNA copy number from infected cells was quantified by qRT PCR and (**C**) infectious virus titer from cell culture supernatants by plaque assay, respectively. Vero E6 cells were pre-treated with 5 or 10 µM montelukast and infected with 0.001 MOI SARS-CoV-2 for 48 hr. (**D**) Virus spike protein expression by western blot analysis, with corresponding relative density of bands. (**E**) Viral RNA copy number from infected cells was quantified by qRT PCR and (**F**) infectious virus titer from cell culture supernatants by plaque assay. *$p < 0.05$; **$p < 0.01$; ***$p < 0.001$; ****$p < 0.0001$; ns-not significant, using one-way ANOVA with Dunnett's multiple comparison test. Error bars represent standard deviation.

The online version of this article includes the following figure supplement(s) for figure 3:

**Figure supplement 1.** Cytotoxicity assay.

**Figure supplement 2.** Saquinavir did not show significant antiviral activity against SARS-CoV-2.

protease, in computational studies (*Abu-Saleh et al., 2020*; *Sharma et al., 2021*). However, Ma and Wang demonstrated that montelukast gives false positive anti-protease activity as it cannot bind the GST-tagged-M$^{pro}$ in thermal shift assay and native mass spectrometry experiments (*Ma and Wang, 2021*). Thus, montelukast may not be an inhibitor for M$^{pro}$ protease.

Viruses employ different strategies to shutdown host translation machinery. In SARS-CoV-2, Nsp1 inhibits translation by binding to the mRNA channel. Here, we show that montelukast binds to Nsp1, rescues the Nsp1-mediated translation inhibition and has antiviral activity against SARS-CoV-2. The rescue of shutdown of host protein synthesis machinery by montelukast seems to contribute toward the antiviral activity of the drug; however, further experiments would be essential to figure out detailed mechanism of its antiviral activity. Overall, our study identifies C-terminal region of Nsp1 as a druggable target and montelukast as a starting point for designing more potent drug molecules against SARS-CoV-2.

# Materials and methods
## Receptor preparation for in silico studies and molecular screening of FDA-approved drugs

The three-dimensional coordinates of C-terminal helices of Nsp1 (Nsp1-C-ter; residue numbers 148–180) were taken from the cryo-EM structure of Nsp1-bound 40S (PDB ID: 6ZOJ). The close

contacts, side chains, and bumps were fixed in Chimera (*Pettersen et al., 2004*). The molecule was minimized using 100 steepest descent steps and ten conjugate gradient steps using AMBERff14SB force field (*Maier et al., 2015*). None of the atoms were fixed during minimization, and charges were assigned using the AMBERff14SB force field on standard residues. The final structure was optimized by Powell method implemented in biopolymer program of SYBYL-X v2.1 (Tripos International, St. Louis, Missouri, 63144, USA).

The FDA-approved drug library was used to screen the drugs toward Nsp1-C-ter. The drug library containing 1,645 compounds was subjected to in silico molecular screening. Three-dimensional structure of (SDF format) compound library was optimized in SYBYL-ligand prep module at default parameters. The single lowest strain energy tautomer for each compound was searched using Surflex in ligand preparation module. Subsequently, the binding pocket for ligands on Nsp1-C-ter was determined by Computed Atlas of Surface Topography of proteins (CASTp) online server (*Tian et al., 2018*). The T151, P153, D156, F157, Q158, N160, K164, H165, S167, T170, R171, E172, L173, R175, and L177 were found to form the binding pocket. Finally, the compound library was screened against 18S rRNA interacting interface of Nsp1-C-ter using the Surflex-dock program, which is available in SYBYL v2.1 (*Jain, 2003*). Twenty conformers were generated for each molecule with 100 maximum rotatable bonds, and top potential molecules were selected based on docking score, which was calculated based on scoring function (flex C-score).

## Nsp1 expression and purification

The gene construct encoding Nsp1 from SARS-CoV-2 in pCDNA 5–3X-Flag-Nsp1 was amplified and sub-cloned into pET28a with N-terminal His-tag (*Schubert et al., 2020*; *Thoms et al., 2020*) using appropriate primers (*Supplementary file 2*). The sub-cloned construct was further used to amplify and clone the C-terminal 28 amino acid deleted construct of Nsp1 (Nsp1ΔC) using appropriate primers (*Supplementary file 2*). Then constructs were transformed into *E. coli* BL-21 DE3 expression system. The secondary cultures were then inoculated with 1% of the primary culture and incubated at 37°C at 180 rpm. At 0.6 O.D., the cultures were induced with 1 mM IPTG at 16°C and 120 rpm for 18 hr. Cells were harvested at 6000 rpm and resuspended in buffer A (50 mM HEPES-KOH pH 7.6, 500 mM KCl, 5 mM $MgCl_2$, 5% Glycerol). Lysis was done by sonicating at 18% amplitude (10 sec on/off cycles for 10 min) and clarified by centrifugation at 12,000 rpm for 30 min. The clear supernatant was then loaded on the Ni-NTA beads (Qiagen) and incubated for 3 hr, and beads were washed using buffer A. The bound protein was eluted with buffer A supplemented with 300 mM imidazole, and purity was analyzed on SDS-PAGE. The fractions containing corresponding protein were concentrated and subjected to size exclusion chromatography on Superdex 200 increase 10/300 column in buffer B (50 mM HEPES-KOH pH 7.6, 150 mM KCl, 5 mM $MgCl_2$, 2% Glycerol and 2 mM DTT). The pure protein fractions were pooled and concentrated between 2 and 8 mg/mL and stored in –80°C for further use.

## Drug-binding assays

### Bio-layer interferometry

To identify the kinetic behavior of the top selected compounds, we performed the label-free binding kinetics of protein and ligands by using bio-layer interferometry. The Ni-NTA sensors were activated by incubating in 10 mM phosphate buffer saline for 10 min. Thereafter, 2 μM of each protein was loaded on the Ni-NTA sensor and a binding response of around 1 nm was obtained. The initial screening of compounds was performed at 20 μM for all in silico selected top hits. The drug molecules that showed binding response of more than 0.2 nm were chosen for further kinetic experiments. The binding kinetics were measured by incubating protein-bound sensors with the increasing ligand concentration (0–25 μM). The data for control sensors (without protein) for each ligand concentration were also collected and subtracted from the response of proteins-bound sensors. The subtracted data was then analyzed by fitting the 1:1 stoichiometric ratio for association and dissociation by applying the global fitting. Three independent experiments were performed to evaluate the steady-state kinetics and calculate $K_D$ values.

### Nanoscale differential scanning fluorometry

In silico identified potential hits were then subjected to evaluate the binding with His-Nsp1 and His-Nsp1ΔC of SARS-CoV-2 protein. 2 μM of each protein was subjected to determine the melting

temperature the in buffer B. The temperature scans ranged from 20 to 90°C with the 1°C/min ramp size using Prometheus NT.48 NanoTemper. Next, the ΔTm was determined in the presence of drug molecules (10 μM) to figure out binding of drug molecules. The top hits were selected for further evaluation in a change of the Tm by incubating with different concentrations of ligand (0–16 μM). The data was analyzed by using ThermControl software.

## Molecular dynamics simulation of C-terminal helices of Nsp1 and drugs-bound complexes

The molecular dynamic simulations of FDA-approved drugs in complex with Nsp1-C-ter were selected based on top binding score using BLI and NanoDSF. The final docked complexes were then prepared for molecular dynamics simulation studies. The systems for molecular dynamics studies were prepared for Nsp1-C-ter alone and their complex with top hits using the Desmond v4.1implemented in Schrodinger-Maestro v11, where steric clashes and side-chain bumps were fixed. These prepared structures were then optimized by GROMOS96 54a7 force field (*Schmid et al., 2011*) and simple point charge water model was used to add the solvent molecules in dodecahedron box with a distance of 1 Å from the surface of protein. Additionally, four sodium ions were added to neutralize the system. The following energy minimization was performed for all the systems with 5000 steps of steepest descent and conjugate gradient algorithms with threshold energy of 100 kcal/mol. The systems were then equilibrated in two phases, first is isothermal-isochoric equilibration, where constant number, volume, and temperature (NVT) was equilibrated for 100 ps, and the temperature of the system was monitored for all constants. In second phase, isothermal-isobaric equilibration was performed where number of particles, pressure, and temperature (NPT) was equilibrated for 100 ps. After successful equilibration of the system, final molecular dynamic runs were performed for 500 ns in three replicas with 2 femtoseconds of time steps. The RMSD, RMSF, and three-dimensional coordinates for all atoms of protein and ligands were extracted to analyze the molecular dynamics runs.

## Binding energy calculation

The binding energy for protein and ligands were calculated by applying the gmx_Molecular Mechanic and Poisson-Boltzmann Surface Area (gmx-MMPBSA) (*Valdés-Tresanco et al., 2021*). Two subsequent 500 ns runs from MD simulations were further subjected to perform the gmx_MMPBSA by using AmberTools21. The binding energy was decomposed into free binding energy for drug molecules for 5000 frames. This binding energy calculation quantitatively provides in silico biomolecular interaction between selected ligands and target protein. This binding energy mainly constitutes the polar solvation energy, non-polar solvation energy and potential energy. The free binding energy ($\Delta G_{binding}$) of the ligand was calculated by the following equation:

$$\Delta G_{\text{binding}} = (G_{\text{complex}}) - G_{\text{receptor}} - (G_{\text{ligand}})$$

Where $\Delta G_{complex}$ describes the Gibbs free energy of the complex, $G_{receptor}$ and $G_{lignad}$ are total energy of protein and ligand, respectively.

## Luciferase-based assay: translation inhibition and rescue experiments

The luciferase based reporter assay was used to evaluate the target-specific action of drug molecules. HEK293 cells were transfected with 100 ng/well of pGL3-Fluc plasmid using Lipofectamine 2000 (Thermo Fisher Scientific) according to the manufacturer's protocol at around 75–90% confluency in a 96 well plate. The plasmid expressing Nsp1 protein (pcDNA 3.1-Nsp1) was co-transfected at 100 ng/well concentration. The transfection was performed in the presence of drugs montelukast and saquinavir at different concentrations. The cells were lysed 24 hr post-transfection, and luciferase activity was measured by using Luciferase Reporter assay system (Promega Corporation) in the GLoMax Explorer system (Promega Corporation).

The expression level of *FLuc* was measured, keeping Glyceraldehyde 3-phosphate dehydrogenase (GAPDH) as the control. Total RNA from all conditions was isolated using the TRIzol as per the user manual protocol. 0.5 μg of total RNA was used as a template for cDNA synthesis (RevertAid First Strand cDNA synthesis kit using manufacturer's protocol), which was further used as template

to quantitate FLuc and GAPDH expression in the presence of appropriate primers as mentioned in *Supplementary file 2*. The relative Ct values were monitored in the three replicates and relative fold change in expression was calculated. The significance of the data was monitored by applying the unpaired t-test through assuming Gaussian distribution parametric test by defining the statistical significance $p < 0.5$.

To evaluate the total viral copy number, RNA from SARS-CoV-2 infected cells was isolated using TRIzol as per manufacturer's instructions, and equal amount of RNA used to determine the viral load using AgPath-ID One-Step RT-PCR kit (AM1005, Applied Biosystems). The primers and probes against SARS CoV-2 N-1 gene used are mentioned in *Supplementary file 1*. A standard curve was made using SARS-CoV-2 genomic RNA standards, which was used to determine viral copy number from Ct values.

## Cells and virus

The following cell lines were used in this study, namely, HEK293 (ATCC), HEK293T-ACE2 (HEK293T cells stably expressing human angiotensin-converting enzyme 2) (BEI Resources NR-52511, NIAID, NIH. RRID: CVCL_A7UK) and Vero-E6 cells (CRL-1586, ATCC, RRID: CVCL_0574). The authenticity of HEK293T-ACE2 and Vero-E6 cell lines was confirmed by Certificate of Analysis from their respective sources. HEK293T-ACE2 are human embryonic kidney 293T cells that express the human ACE2 receptor, which is required for SARS-CoV-2 entry. HEK293T-ACE2 and Vero E6 cells are of human and primate origin respectively, and express ACE2 receptor. All cell lines tested negative for mycoplasma contamination. Cells were cultured in complete media prepared using Dulbecco's modified Eagle medium (12100–038, Gibco) supplemented with 10% HI-FBS (16140–071, Gibco), 100 U/mL Penicillin-Streptomycin (15140122, Gibco) and GlutaMAX (35050–061, Gibco).

SARS-CoV2 (Isolate Hong Kong/VM20001061/2020, NR-52282, BEI Resources, NIAID, NIH) was propagated and quantified by plaque assay in Vero-E6 cells as described before (*Case et al., 2020*).

## Cytotoxicity assay

HEK293T-ACE2 cells were seeded in 0.1 mg/mL poly-L-lysine (P9155-5MG, Sigma-Aldrich) coated 96-well plate to reach 70–80% confluency after 24 hr. Vero-E6 cells were seeded in a regular 96 well plate to reach similar confluency. Cells were treated with 5, 10, and 20 µM montelukast or saquinavir in triplicates and incubated at 37°C/5% $CO_2$. After 48 hr, cytotoxicity was measured using AlamarBlue Cell Viability Reagent (DAL 1025, Thermo Fisher Scientific) as per manufacturer's instructions.

## Western blot

Cells were washed gently with 1× warm PBS (162528, MP Biomedicals), lysed using 1× Laemmli buffer (1610747, BIO-RAD), and heated at 95°C before loading on to a 10% SDS-PAGE gel. Separated proteins were transferred onto a PVDF membrane (IPVH00010, Immobilon-P; Merck) and incubated for 2 hr with blocking buffer containing 5% Skimmed milk (70166, Sigma-Aldrich) in PBST (1× PBS containing 0.05% Tween 20 (P1379, Sigma-Aldrich)) for 2 hr at RT (room temperature). The blots were then probed with SARS-CoV-2 spike antibody (NR-52947, BEI Resources, NIAID, NIH) in blocking buffer for 12 hr at 4°C, followed by secondary Goat Anti-Rabbit IgG antibody (ab6721, Abcam, RRID:AB_955447) incubation for 2 hr. Proteins were detected using Clarity Western ECL Substrate (1705061, BIO-RAD). Actin was labeled using antibody against beta-actin [AC-15] (HRP) (ab49900, Abcam, RRID: AB_867494). Relative intensity of bands was quantified using imagej/Fiji.

## Virus infection

HEK293T-ACE2 cells were seeded in poly-L-lysine coated 24-well plate to reach 80% confluency at the time of infection. Vero-E6 cells were seeded in a regular 24 well plate to reach similar confluency. Cells, in quadruplicates, were first pre-treated with 5 and 10 µM concentrations of montelukast sodium hydrate (PHR1603, Merck) or saquinavir mesylate (1609829, Merck) for 3 hr in complete media, washed and infected with 0.1 MOI (HEK ACE2) or 0.001 MOI (Vero-E6 cells) SARS-CoV-2. After 48 hr, cell culture supernatants were collected for plaque assay, and cells were harvested for western blot analysis or processed for total RNA extraction using TRIzol (15596018, Thermo Fisher Scientific). The drugs were present in the media for the entire duration of the experiment.

## Plaque assay

Infectious virus particles from cell culture supernatants were quantified by plaque assay. Briefly, Vero-E6 cells were seeded in 12-well cell culture dishes, and once confluent, cells were washed with

warm PBS and incubated with dilutions of cell culture supernatants in 100 µL complete DMEM for 1 hr at 37°C/5% $CO_2$. The virus inoculum was then removed, and cells overlaid with 0.6% Avicel (RC-591, Dupont) in DMEM containing 2% HI-FBS. After 48 hr incubation, cells were fixed with 4% paraformaldehyde, and crystal violet (C6158, Merck) staining was done to visualize the plaques.

Plasmids pLVX-EF1alpha-SARS-CoV-2-nsp1-2xStrep-IRES-Puro expressing SARS-CoV-2 NSP1 was a kind gift from Prof. Nevan Krogan (*Gordon et al., 2020*). Other plasmids used in this study include Plasmids pRL-TK (mammalian vector for weak constitutive expression of wild-type Renilla luciferase), pGL4 (mammalian vector expressing firefly luciferase), pIFN-β Luc (IFN beta promoter-driven firefly luciferase reporter). The plasmid pMTB242 pcDNA5 FRT-TO-3xFLAG-3C-Nsp1_SARS2 was a kind gift from Prof. Ronald Beckmann.

## Supporting Information

Supporting information contains four figures and two supplementary files.

## Acknowledgements

This work was supported by Intermediate Fellowship from DBT-Wellcome Trust India Alliance to TH (IA/I/17/2/503313). TH also thanks SERB for funds released under IRPHA (COVID-19 Life Sciences; File Number:IPA/2020/000094). ST acknowledges funding from DBT-BIRAC grant (BT/CS0007/CS/02/20) and DBT-Wellcome Trust India Alliance Intermediate Fellowship (IA/I/18/1/503613). We acknowledge Swarnajayanti Fellowship from DST to SME (SB/SJF/2020-21/18).

## Additional information

### Funding

| Funder | Grant reference number | Author |
|---|---|---|
| Wellcome Trust/DBT India Alliance | IA/I/17/2/503313 | Tanweer Hussain |
| IRPHA | IPA/2020/000094 | Tanweer Hussain |
| Wellcome Trust/DBT India Alliance | IA/I/18/1/503613 | Shashank Tripathi |
| Swarnajayanti Fellowship | SB/SJF/2020-21/18 | Sandeep M Eswarappa |
| Biotechnology Industry Research Assistance Council | BT/CS0007/CS/02/20 | Shashank Tripathi |

The funders had no role in study design, data collection and interpretation, or the decision to submit the work for publication.

### Author contributions

Mohammad Afsar, Formal analysis, Investigation, Methodology, Validation, Writing – original draft, Writing - review and editing; Rohan Narayan, Md Noor Akhtar, Formal analysis, Investigation, Methodology, Writing - review and editing; Deepakash Das, Investigation, Methodology, Writing - review and editing; Huma Rahil, Investigation, Methodology; Santhosh Kambaiah Nagaraj, Investigation; Sandeep M Eswarappa, Shashank Tripathi, Funding acquisition, Supervision, Writing – original draft; Tanweer Hussain, Conceptualization, Funding acquisition, Resources, Supervision, Writing – original draft, Writing - review and editing

### Author ORCIDs

Md Noor Akhtar http://orcid.org/0000-0002-4669-1543
Sandeep M Eswarappa http://orcid.org/0000-0002-7903-5198
Tanweer Hussain http://orcid.org/0000-0003-4735-2380

### Decision letter and Author response

Decision letter https://doi.org/10.7554/eLife.74877.sa1

Author response https://doi.org/10.7554/eLife.74877.sa2

## Additional files

### Supplementary files
- Supplementary file 1. Top hits of FDA-approved drugs upon screening against Nsp1-C-ter.
- Supplementary file 2. Primers /oligos used in this study.
- Transparent reporting form
- Source data 1. Data for making figures.

### Data availability
All data generated or analyzed during this study are included in the manuscript and supporting file.

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
