## [Editor Report]

This study reports on the repurposing of Montelukast, an FDA-approved drug, with Nsp-1. The Non-structural protein (Nsp)-1 from SARS-CoV2 mimics the binding mode of eukaryotic initiation factor 3 (eIF3j) to the mRNA entry tunnel of the 40S ribosomal subunits and blocks the entry of mRNA, which shuts down host protein synthesis. As a result, the host immune function is suppressed. This makes Nsp-1 an attractive target for therapeutic intervention. In the revised manuscript the authors' claims are supported using Biophysical and cellular assays. They also rationalize their findings using molecular dynamics simulations.

---

## [Decision Letter]

**Decision letter after peer review:**

Thank you for submitting your article "Drug targeting Nsp1-ribosomal complex shows antiviral activity against SARS-CoV-2" for consideration by *eLife*. Your article has been reviewed by 3 peer reviewers, including Shozeb Haider as Reviewing Editor and Reviewer #1, and the evaluation has been overseen by Mone Zaidi as the Senior Editor.

Essential revisions:

Please see the recommendation for the authors below.

*Reviewer #1 (Recommendations for the authors):*

The short report is well written and the experimental data supports the findings. The only shortcoming (though not a limitation) is the length of the MD simulation. They should be at least be 500ns to remove any doubts of the system still equilibrating.

There are too many figures for a short report. The drug screening data (in Figure 1C D E F) should be moved to SI and Figure 2 merged with Figure 1.

*Reviewer #2 (Recommendations for the authors):*

1. In my understanding the c-terminal region on Nsp1 is disordered in solution and folds only upon binding. Thus, it might be not a solid target for virtual screening and the proposed binding modes might be greatly affected by the flexibility of the region. In this respect the MD simulations might help to confirm the binding mode, but the simulations reported are too short (200ns) to be really conclusive.

2. The calculated binding free energies (-76 and -72 kJ/mol) are far better than the reported 10 and 7 μM binding affinities. Even accounting for the well-known lack of accuracy of MM-PBSA, the discrepancy is too large. Thus, a more predictive method, such as alchemical free energy calculations, should be used to confirm that the proposed binding modes are indeed the correct ones.

3. The experimental binding affinities to Nsp1 are quite low. Thus, the observed in vivo effects might be due to the targeting of a different viral protein, perhaps even the reported anti-protease activity.

Overall, there are many important points to clarify and thus the conclusions are not currently supported by the data.

*Reviewer #3 (Recommendations for the authors):*

I recommend that the authors change "in vivo" models to cellular models on page 4, line 84.

The last sentence in the discussion should be changed to stating that montelukast will be an excellent starting point for the development of more potent drugs.

Abstract, lines 35 and 37: change "cov" to "CoV".

Lines 46 and 47: change sentence as montelukast will not help to combat SARS-CoV-2 as the Kd valu is too weak with ca. 10 uM. But one could argue that it is a good starting point for the development of more potent drugs.

I think the authors could improve their manuscript by polishing the English used. Please also adjust the use of the article "the". In addition, sentences could be improved in line 97, lines 136 to 137 (improve first part of the sentence), lines 143 to 144 (change "was" to "were"), lines 155 to 156, line 285 (change to "amino acids 145 to 180"), line 352 and line 381, to name a few. As this is an interesting manuscript, it is worth improving it!

---

## [Author Response]

Reviewer #1 (Recommendations for the authors):The short report is well written and the experimental data supports the findings. The only shortcoming (though not a limitation) is the length of the MD simulation. They should be at least be 500ns to remove any doubts of the system still equilibrating.

We thank the reviewer for suggestion, and we have performed MD simulation runs (three replica’s) for each protein-ligand complex for 500 ns. The RMSD plots and final conformations of the ligand is incorporated in Figure 1. The RMSF plot is also updated in Figure 1 supplement figure 1. The RMSD plot for montelukast shows minimal fluctuations throughout the simulation runs.

There are too many figures for a short report. The drug screening data (in Figure 1C D E F) should be moved to SI and Figure 2 merged with Figure 1.

The figures have been updated accordingly.

Reviewer #2 (Recommendations for the authors):1. In my understanding the c-terminal region on Nsp1 is disordered in solution and folds only upon binding. Thus, it might be not a solid target for virtual screening and the proposed binding modes might be greatly affected by the flexibility of the region. In this respect the MD simulations might help to confirm the binding mode, but the simulations reported are too short (200ns) to be really conclusive.

We thank the reviewer for this suggestion. We now report simulation runs of 500 ns for Nsp1-C-ter bound with drugs to provide insights into the binding mode. The molecular dynamic runs show the formation of stable complexes.

Regarding the C-terminal region of Nsp1, we performed secondary structure prediction of C-terminal region of Nsp1 by using the PSIPRED, JNET and JURY-OF-EXPERTS PREDICTION online tools. These tools predicted the formation of two α-helices in the C-terminal region of Nsp1.

**Author response image 1. sa2fig1:** Secondary structure prediction of C terminal region of Nsp1 shows the adaptation of α-helical conformation. This region forms two helices which are connected by loop region.

Thus, based on above predictions, it is likely that in solution the C-terminal region of Nsp1 forms α-helices.

The next question is whether the two helices are bound together in the same fashion as observed seen in the mRNA channel? We argue that the C-terminal region of Nsp1 must attain this conformation before entering the mRNA channel because the mRNA channel is too narrow to allow large conformational changes. Secondly, the presence of hydrophobic interactions between the F157, P153, W161 from helix1 and V169, L173, L177 from helix2 indicate the likelihood of attaining this conformation in solution (as the hydrophobic residue would be shielded away from water).

**Author response image 2. sa2fig2:** Hydrophobic amino acids in C terminal helices of Nsp1 shows the presence of hydrophobic interactions between the two helices.

Thirdly, the molecular dynamic runs for 500 ns show the formation of stable complexes.

While we agree with the reviewer that there may some flexibility in the C-terminal region of Nsp1 in solution; however, the ribosome-bound structure (also only available structure) was a good starting structure for *in silico* screening, which we were able to validate with in vitro ligand binding assays and cell based assays.

2. The calculated binding free energies (-76 and -72 kJ/mol) are far better than the reported 10 and 7 μM binding affinities. Even accounting for the well-known lack of accuracy of MM-PBSA, the discrepancy is too large. Thus, a more predictive method, such as alchemical free energy calculations, should be used to confirm that the proposed binding modes are indeed the correct ones.

We thank reviewer for this suggestion to use more accurate method for calculating binding energy of protein-ligand complexes. We have performed End-State Free Energy Calculations for initial 5000 frames using the Ambertools20 to calculate binding energy. We found that montelukast binds with -95.8±13.3 kJ/mol (-22.9±3.2 kcal/mol) and saquinavir binds with -42.7±5.2 kJ/mol (-10.2±1.2 kcal/mol). These free energy calculations suggest stable binding of montelukast compared to saquinavir.

We note that the discrepancy is large between the calculated binding free energies and binding affinities. This may be because the *in silico* binding free energies are calculated with only the C-terminal region of Nsp1 only while the in vitro binding affinities are obtained from experiments with full Nsp1. In the absence of structure of full Nsp1, it is difficult to ascertain the position of C-terminal region with respect to the N-terminal domain. It is likely that in the context of full Nsp1, the C-terminal region is not as accessible (when compared to only ‘C-terminal region of Nsp1’ used for calculating binding free energies).

3. The experimental binding affinities to Nsp1 are quite low. Thus, the observed in vivo effects might be due to the targeting of a different viral protein, perhaps even the reported anti-protease activity.

We have performed cell-based translational rescue of luciferase activity in the presence of drugs in HEK293 cells when co-transfected with Nsp1. So, the only viral protein co-transfected in this experiment was Nsp1. Here we observed that co-transfection of Nsp1 decreased the luciferase activity by almost half, which is restored by the increasing amount of montelukast (Figure 2A). This experiment shows that montelukast targets Nsp1-ribosome interaction to restore translation. SARS-CoV-2 evades immune response by blocking the translation of antiviral defence factors using Nsp1. Restoring host translation would aid in viral clearance as it allows synthesis of major immune effectors like IFN-β, IFN-l1, and interleukin-8, retinoic acid–inducible gene I (Thoms et al., 2020, Science *369*, 1249-1255. http://www.ncbi.nlm.nih.gov/pubmed/32680882).

We agree that the binding affinity of montelukast to Nsp1 is low. Hence we have now modified our statement to say that montelukast would be a good starting point for designing more potent drug molecules against SARS-CoV-2 (Pg 2 lines 46-47 and Pg 7, line 207-208).

Regarding the reported anti-protease activity of montelukast, Ma and Wang showed that montelukast gives false positive anti-protease activity and it cannot bind the GST-tagged-M^pro^ in thermal shift assay and native mass spectrometry experiments. Hence, they concluded that montelukast is not SARS-CoV-2 main protease inhibitor (Ma and Wang, Proc Natl Acad Sci, 2021; doi: 10.1073/pnas.2024420118). We have discussed this work in discussion (Pg 7, line 198-200).

Hence, the observed antiviral effects of montelukast appears to be because of targeting of the C-terminal helices of Nsp1.

Overall, there are many important points to clarify and thus the conclusions are not currently supported by the data.Reviewer #3 (Recommendations for the authors):I recommend that the authors change "in vivo" models to cellular models on page 4, line 84.

We thank reviewer for this valuable suggestion. We have made this change throughout the manuscript.

The last sentence in the discussion should be changed to stating that montelukast will be an excellent starting point for the development of more potent drugs.

We have made this change in the last sentence of the discussion as suggested. Further, we have modified the last sentence of the discussion.

Abstract, lines 35 and 37: change "cov" to "CoV".

This is corrected in the revised manuscript.

Lines 46 and 47: change sentence as montelukast will not help to combat SARS-CoV-2 as the Kd valu is too weak with ca. 10 uM. But one could argue that it is a good starting point for the development of more potent drugs.

We have made this change in the revised manuscript.

I think the authors could improve their manuscript by polishing the English used. Please also adjust the use of the article "the". In addition, sentences could be improved in line 97, lines 136 to 137 (improve first part of the sentence), lines 143 to 144 (change "was" to "were"), lines 155 to 156, line 285 (change to "amino acids 145 to 180"), line 352 and line 381, to name a few. As this is an interesting manuscript, it is worth improving it!

We have rewritten portion to improve the English used. Also, we have made corrected the typos throughout the manuscript.